# Predisposing and Precipitating Factors in Epstein–Barr Virus-Caused Myalgic Encephalomyelitis/Chronic Fatigue Syndrome

**DOI:** 10.3390/microorganisms13040702

**Published:** 2025-03-21

**Authors:** Leonard A. Jason, Ben Z. Katz

**Affiliations:** 1Center for Community Research, DePaul University, Chicago, IL 60614, USA; 2Feinberg School of Medicine, Ann and Robert H. Lurie Children’s Hospital of Chicago, Northwestern University, Chicago, IL 60208, USA; bkatz@luriechildrens.org

**Keywords:** prospective, ME/CFS, risk factors, cytokines

## Abstract

Long COVID following SARS-CoV-2 and Myalgic Encephalomyelitis/Chronic Fatigue Syndrome (ME/CFS) following infectious mononucleosis (IM) are two examples of post-viral syndromes. The identification of risk factors predisposing patients to developing and maintaining post-infectious syndromes may help uncover their underlying mechanisms. The majority of patients with ME/CFS report infectious illnesses before the onset of ME/CFS, with 30% of cases of ME/CFS due to IM caused by the Epstein–Barr virus. After developing IM, one study found 11% of adults had ME/CFS at 6 months and 9% had ME/CFS at 1 year. Another study of adolescents found 13% and 7% with ME/CFS at 6 and 12 months following IM, respectively. However, it is unclear which variables are potential risk factors contributing to the development and maintenance of ME/CFS following IM, because few prospective studies have collected baseline data before the onset of the triggering illness. The current article provides an overview of a study that included pre-illness predictors of ME/CFS development following IM in a diverse group of college students who were enrolled before the onset of IM. Our data set included an ethnically and sociodemographically diverse group of young adult students, and we were able to longitudinally follow these youths over time to better understand the risk factors associated with the pathophysiology of ME/CFS. General screens of health and psychological well-being, as well as blood samples, were obtained at three stages of the study (Stage 1—Baseline—when the students were well, at least 6 weeks before the student developed IM; Stage 2—within 6 weeks following the diagnosis of IM, and Stage 3—six months after IM, when they had either developed ME/CFS or recovered). We focused on the risk factors for new cases of ME/CFS following IM and found factors both at baseline (Stage 1) and at the time of IM (Stage 2) that predicted nonrecovery. We are now collecting seven-year follow-up data on this sample, as well as including cases of long COVID. The lessons learned in this prospective study are reviewed.

## 1. Predisposing and Precipitating Factors in Epstein–Barr Virus-Caused ME/CFS

Epstein–Barr virus (EBV) causes almost all cases of heterophile antibody positive Infectious Mononucleosis (IM), and the heterophile antibody test is positive in about 90% of young adults who develop IM (as reviewed in [1]). Several studies have attempted to better define the relationship between EBV and Myalgic Encephalomyelitis/Chronic Fatigue Syndrome (ME/CFS). For example, White et al. [2] assessed patients with IM or an upper respiratory tract infection for the development of fatigue and/or ME/CFS. Nine percent of subjects with IM, due to EBV, were fatigued and complained of excessive sleeping at 6 months, compared with none who had a previous upper respiratory tract infection. Hickie et al. [3] showed an 11% rate of ME/CFS 6 months following IM as well as following 2 other similar systemic infections. In another study, following a diagnosis of IM among youths, the incidence of ME/CFS at 6, 12, and 24 months was 13%, 7%, and 4%, respectively [4].

Most patients with IM improve over time, whereas those individuals who develop ME/CFS remain impaired [5]. There are likely a host of predisposing, precipitating, and perpetuating factors for the development of ME/CFS, which might be similar or different to risk factors identified for developing long COVID (e.g., SARS-CoV-2 viremia, Epstein–Barr viremia, specific autoantibodies, type II diabetes, obesity, elevated blood pressure, chronic lung disease, and depression [6]).

There is evidence of inflammation among those with ME/CFS. Increases in IL-8 in the cerebrospinal fluid of some patients with ME/CFS have been reported, supporting the hypothesis that symptoms may be related to immune dysfunction within the central nervous system [7]. Broderick et al. [8] applied network analysis to cytokines in patients with ME/CFS and healthy controls; outcomes were consistent with attenuated T-cell helper [Th]1 and Th17 immune responses in the presence of a Th2 inflammatory milieu. Thus ME/CFS following IM might be an extension of the known autoimmune phenomena that can accompany primary EBV infection [9].

Multiple studies in recent years have reported detectable changes in metabolic pathways related to energy production, amino acids, nucleotides, nitrogen, lipids, and neurotransmitters in patients with ME/CFS [10,11,12,13]. Other studies have indicated an increased risk of ME/CFS among close relatives of index patients, suggesting a heritable component [14]. People with ME/CFS with positive ME/CFS family histories are more likely to have gastrointestinal symptoms than those with ME/CFS without those family histories [15]. Guo et al. [16] and Xiong et al. [17] have found disruptions in the gastrointestinal microbiome among patients with ME/CFS, and we have found that GI symptoms predispose to the development of severe ME/CFS following IM [18]. Furthermore, the disruptions in the gastrointestinal microbiome described in patients with ME/CFS are associated with an increase in pro-inflammatory and a reduction in anti-inflammatory species [19].

There are a range of other areas of impaired biological functioning among those with ME/CFS and long COVID, including deficits in cerebral blood flow [20]. Thapaliya et al. [21] found significantly larger volumes of the pons, the superior cerebellar peduncle, and the brainstem in both patients with long COVID and ME/CFS, and inflammation could be causing varied deficits in brain function in these patients (e.g., brain fog). “Diffuse white-matter disease” might also contribute to the cognitive difficulties seen in patients with ME/CFS and long COVID [22]. Inflammatory immune system response to the virus, injury to blood vessels, and/or lack of oxygen to the brain may account for these changes. Finally, microglia in the white matter of the brain, which are responsible for pruning the connections between neurons to improve neural circuits, could be damaged by cytokines (e.g., CCL11) and consequently reduce the generation of new nerve cells, affect memory formation [23], and lead the neurological symptoms of long COVID and/or ME/CFS. Zinn and Jason [24] explored the role of the cortical autonomic network involved in higher-order control of autonomic nervous system functioning in patients with ME/CFS and healthy controls under resting-state quantitative electroencephalographic (qEEG) scalp recordings and found evidence for reduced higher-order homeostatic regulation and adaptability in ME/CFS.

Almost all studies reviewed above examined patients after becoming ill, with some patients being infected with EBV or SARS-CoV-2. A key question is why some people remain ill after a viral infection, while others recover. Prospective studies of individuals before infection may yield critical insights about conditions that predispose to long-term illness. For the past decade, the authors have been working to identify biological profiles and behavioral domains that predispose and precipitate ME/CFS, and our work is described below.

## 2. Prospective Study

From 2014 through 2018, data were collected from 4501 demographically diverse college students at least 6 weeks before the development of IM (baseline, Stage 1). They were then followed for the development of IM (Stage 2—diagnosed via a positive monospot or specific Epstein–Barr virus serologies [a positive viral capsid antigen {VCA} immunoglobulin {Ig} M or a positive VCA IgG with a negative Epstein–Barr nuclear antigen antibody]). Six months later (Stage 3), they were assessed for the development of ME/CFS or recovery; for further details see Jason et al. [25]. Individuals were examined by a physician in Stage 3 and this information as well as self-report questionnaires was used to designate whether the person had ME/CFS. We stored biological samples from each stage of the study, including pre-illness serum and plasma, as well as viable white blood cells from Stages 2 and 3.

Figure 1 shows that 238 of the 4501 students (5.3%) developed IM. Six months later, 55 of the 238 met the criteria for ME/CFS, and 157 were asymptomatic [25]. In all, 67 of the 157 asymptomatic students served as recovered controls. Students with “severe” ME/CFS (those who met > 1 set of criteria for ME/CFS) were compared to students who met a single set of criteria (“moderate” ME/CFS) and to those who recovered 6 months following IM (See Figure 1). We did not find any significant differences between those who developed ME/CFS versus those who recovered, on pre-illness baseline differences in stress, coping, anxiety, or depression. We did find baseline pre-illness complaints of fatigue and deficiencies in IL-5 and IL-13 in the group that went on to develop severe ME/CFS versus those who recovered. Deficiencies in IL-5 and IL-13 before contracting IM may influence the immune response once the virus is contracted. For example, there is evidence in human and mouse models that IL-5 and IL-13 contribute to the pathology associated with ulcerative colitis.

## 3. Baseline Network Analysis

In a network analysis study, we next examined groups of cytokines within each condition before developing IM. Figure 2 and Figure 3 show intercommunication in the pre-illness immune systems of the severe ME/CFS group and the recovered controls. It is evident that when compared to the groupings of cytokines in the recovered controls, the cytokines of those who went on to develop severe ME/CFS are highly clustered [26]. This network analysis suggests that the baseline deficiencies of IL5 and IL13 in the severe ME/CFS group may have led to the clustering of cytokines seen at baseline in the participants who developed severe ME/CFS following IM compared to those who recovered. More differentiated cytokine networks were seen for the recovered controls at baseline. In general, as with findings from Sorenson et al. [27], we found a different pattern of cytokine activation in control subjects versus subjects with post-viral fatigue, both before developing IM and after [26]. In the ME/CFS sample, the in silico-modeled cytokine-association patterns were also more interwoven, with less grouping into functional categorizations than in the healthy controls. The implication is that pathway activation is less discrete and more reflective of an ill-orchestrated immunologic response in those who developed ME/CFS. The differences in the pattern associations between these two samples were statistically significant, providing support for an immunologic pathogenic process.

## 4. Predicting ME/CFS

We next examined other predictors (i.e., other pre-illness variables as well as variables present at the onset of IM) of those who developed moderate and severe ME/CFS following IM. Multiple data points included seven self-report questionnaires, physical examination findings, the severity of the mononucleosis scale [28], and cytokine analyses. Two random forest classification analyses were conducted to predict the development of ME/CFS [18]. Patients with stomach pain, bloating, and symptoms of an irritable bowel at pre-illness baseline, low levels of IL-13 and/or IL-5 at pre-illness baseline (previously discussed), and severe gastrointestinal symptoms at the time they contracted mononucleosis had a nearly 80% chance of developing severe ME/CFS six months following IM.

## 5. Baseline Metabolic Pathways

Metabolomics has allowed investigators to better understand which metabolic pathways are dysregulated in several different diseases. In our next study, Jason et al. [29] examined baseline metabolomics levels among those with severe ME/CFS versus controls. A series of binary logistic regressions were conducted to classify the severe ME/CFS and recovered groups. Significant differences were observed for the following metabolites: S-adenosyl-L-methionine (part of one-carbon metabolism and is a methyl donor for epigenetic regulation), glutathione (part of glutathione metabolism), cysteine (an amino acid that participates in a variety of pathways, including glutathione metabolism), thiamine (modified and used as a cofactor in several TCA cycle enzymes), and N-acetyl-alanine (may have a role in protein signaling and post-translational modifications). The models produced correctly classified those with severe ME/CFS from recovered controls with an accuracy of 97%, sensitivity of 94%, and specificity of 100%. We thus identified potentially dysregulated pre-illness pathways that are essential for proliferating cells, particularly during a pro-inflammatory immune response, and are thus consistent with the irregularities in cytokines seen in our prior studies (see Table 1).

## 6. Long COVID

Our studies of ME/CFS following IM began before the onset of the COVID-19 pandemic. We are now collecting 7-year follow-up data from our original sample and are also gathering information regarding whether the students had been infected with SARS-CoV-2 in the interim and whether or not they recovered. We are also studying a new group of college students who did and did not recover from SARS-CoV-2 and will compare and contrast the mechanisms of nonrecovery from IM and COVID-19. These participants with SARS-CoV-2 infection were recruited in 2023–2024 by social media sources and from posters and recruitment efforts at local universities. Participants recruited were provided a medical evaluation, and that along with the self-report questionnaires helped determine whether a person had recovered or not from SARS-CoV-2. Those classified as long COVID had continuous, relapsing, or remitting symptoms affecting one or more organ symptoms for at least three months following acute SARS-CoV-2 infection [30]. Matching occurred on sex, race, ethnicity, and age for those with ME/CFS following IM and those with long COVID, and the controls who recovered from mono and those who recovered from SARS-CoV-2.

## 7. Discussion

Insights from our prospective study and others can help improve efforts to understand ME/CFS pathophysiology, early diagnosis, and prognosis. We found that at Stage 1, those fated to develop ME/CFS 6 months following IM had low levels of IL-5 and IL-13 [25]. IL-5 enhances the production of B1 cells which are anti-inflammatory (impaired B1 cells have been found in multiple sclerosis, systemic lupus erythematosus, and rheumatoid arthritis), and IL-13 has anti-inflammatory properties. Jason et al. [26] also found that students who failed to recover from IM had more restricted, inflexible cytokine networks whereas those who recovered had more flexible, less interconnected networks. Network analysis suggests that IL-5 and IL-13 influence other cytokines at baseline in the participants who are fated to develop ME/CFS following IM compared to those who recover. Both IL-5 and IL-13 are critical signaling proteins for eosinophil recruitment and production [31]; deficiencies in these cytokines before contracting IM may cause the immune system to lack the breadth or suppleness needed to cope successfully with primary EBV infection. Thus, the wider range of resources available to those with looser and broader immune networks may have helped them recover. In addition, the ME/CFS group displayed a “lack of involvement” in networks that are effective in fighting off EBV, as found by Loebel et al. [32]. To summarize, those who go on to develop post-viral fatigue after IM had the densest networks of cytokines pre-illness, and it is cytokine networks that are thought to drive disease processes such as inflammatory bowel disease [33].

A consistent theme in our work has been differentiating patients who have what we have termed severe ME/CFS from those with moderate ME/CFS. Those with severe ME/CFS meet more than just the Fukuda criteria, which are less specific than other criteria for diagnosing ME/CFS. Those in the severe group in contrast to controls had saliva biomarkers of fatigue [34], more dense pre-illness interconnected cytokine networks [26], and more gastrointestinal distress and autonomic symptoms within 2 months of their developing IM, along with several immune [18], and metabolomic pre-illness biomarkers [29].

We have found many benefits of using multiple methods in assessing our participants. We provided our participants with comprehensive physician exams to assess for multiple orthostatic indicators (pulse, blood pressure, syncope); testing of plasma cytokines, lymphocyte counts, natural killer cell counts and activity, and T-cell subsets; autoantibodies; metabolomic analysis; and saliva. We also developed the severity of the Mononucleosis Scale to assess the severity of IM. In addition, we collected family history of ME/CFS and other diseases and validated self-report ratings to measure fatigue, symptoms, limitations, and psychological variables. We benefited from working with a multidisciplinary team of investigators, including those from infectious disease medicine, clinical psychology, computer science, immunology, metabolomics, and neurology. We also employed multiple types of statistics including network analysis, data mining, machine learning, logistic regression, and receiver operating characteristic curves.

In working with ME/CFS, we also documented the presence of ME/CFS by using validated questionnaires such as the DePaul Symptom Questionnaire (DSQ), conducted medical and psychological examinations, and excluded other diagnoses such as anemia, hypothyroidism, and depression. We have learned it is important to use methods that can rely on more than just asking patients whether or not they have ME/CFS or long COVID. In adult and pediatric community-based studies, 90 to 95% of participants are not even aware they have ME/CFS [35,36]. In a study with a sample of 465 individuals with long COVID, of respondents who reported that they had ME/CFS, 29% did not meet the criteria for ME/CFS [37]. Relying purely on self-reporting of ME/CFS and long COVID status can result in over-diagnosis and under-diagnosis.

Because fatigue affects about 25% of the population, we have found it is inadequate to merely assess whether a patient has experienced the occurrence (yes versus no) of fatigue or other common somatic symptoms. These types of measures will not differentiate ME/CFS from psychiatric conditions such as Major Depressive Disorder [38]. Thus, we also measure the frequency and severity of each symptom, using the DePaul Symptom Questionnaire, to better differentiate cases of ME/CFS from controls [39].

There is a clear need to better understand similarities and differences between varieties of post-viral illnesses, such as long COVID and ME/CFS following IM. For example, in one study using the DePaul Symptom Questionnaire, those with long COVID had similar symptom scores to patients with ME/CFS [40]. A year later, five symptoms improved significantly for those patients with long COVID including fatigue, post-exertional malaise, brain fog, irritable bowel symptoms, and feeling unsteady. In contrast, there were no significant symptom improvements for the patients with ME/CFS a year later. Using another data set with the DePaul Symptom Questionnaire, McGarrigle et al. [41] examined differences between those with ME/CFS and long COVID and found that “Cold limbs” and “Flu-like symptoms” were significantly more likely to occur in the ME/CFS group. Finally, in another study using the DePaul Symptom Questionnaire, Hua et al. [42] built predictive models based on a random forest algorithm analysis using the participants’ symptoms from the initial weeks of COVID-19 infection to predict if the participants would go on to later meet the criteria for ME/CFS. Early symptoms, particularly those of post-exertional malaise, predicted the development of ME/CFS with an accuracy of 95%.

There are probably similarities in pathophysiology between ME/CFS due to IM and long COVID. After EVB infection, Müller-Durovic et al. [43] found the viral protein, EBV-encoded transactivator EBNA2, in cooperation with the host B cell transcription factor EBFI, drove induction of indoleamine 2,3-dioxygenase 1 (IDO1), the first and rate-limiting enzyme of the kynurenine pathway. This same pathway may be involved in the development of long COVID as well [44]. Following acute COVID-19 infection, some patients experience a strong inflammatory response, with increased levels of interferon-y (IFN-y), interferon-p (IFN-P), interleukin-6 (IL-6), and tumor necrosis factor-alpha (TNF-a). These cytokines activate the enzyme IDO1, causing the first step of tryptophan breakdown through the kynurenine pathway, leading to the production of several neurotoxic and immunosuppressive metabolites, which might be responsible for some of the symptoms of post-viral illnesses such as brain fog. Ruffieux et al. [45] studied individuals infected with SARS-CoV-2 a year after disease onset. They found that patients had a metabolic signature characterized by increased expression of intermediates from the kynurenine pathway and depletion of the upstream amino acid tryptophan. The authors suggest that abnormal levels of kynurenine-pathway intermediates, coupled with the significant reduction in serotonin, contribute to fatigue, weakness, and chronic pain of long COVID.

Since the start of the SARS-CoV-2 pandemic, most research funding has been directed at long COVID rather than ME/CFS. This might be due to scientists and government funders perceiving long COVID to have a clear, specific trigger whereas ME/CFS does not. However, SARS-CoV-2 has continually evolved, resulting in the emergence of several lineages and variants of concern. Long COVID samples often include different variants (e.g., Alpha, Beta, Gamma, Delta, Omicron) that have different transmission, severity, and immune-evasion properties. The majority of patients with ME/CFS report infectious illnesses before the onset of ME/CFS, with 30% of cases of ME/CFS due to preceding infectious mononucleosis caused by the Epstein–Barr virus (EBV) [46]. Therefore, it is possible to study ME/CFS caused by a single virus, EBV.

A better understanding of the etiology and biomarkers for ME/CFS might lead to treatments. Regrettably, there are no FDA-approved ME/CFS treatments. Consequently, healthcare professionals have prioritized the management of ME/CFS symptoms such as fatigue through energy conservation activities [47], the control of postural tachycardia syndrome [48,49,50], and the management of pain [51], sleep disorders [52], and gastrointestinal dysbiosis [53]. Inflammation, immune dysfunction, neuroinflammation, and mitochondrial dysfunction may play a role in the etiology of post-viral fatigue (e.g., ME/CFS and long COVID) [54], with subsequent developments of therapeutics. Since the majority of these studies have not been replicated and the majority of the tested drugs have not been subjected to large, well-designed, randomized, placebo-controlled trials, it is currently not possible to evaluate the evidence of efficacy. Regrettably, most patients with ME/CFS report they benefited relatively little from most interventions offered to them, and many report that the interventions negatively affected their health [55], and as a consequence, patients report low satisfaction with the medical care they received [56]).

The current study only reports on a few of the current investigations that will occur with this prospective study. Certainly, there are multiple other areas that can be further investigated with this data set involving cognitive, genetic, and sex-related factors. For example, Pipper et al. [57] recently found that females with ME/CFS had higher levels of 11-deoxycortisol, 17α-hydroxyprogesterone, and progesterone levels versus controls, whereas males with ME/CFS had lower circulating levels of cortisol and corticosterone, and higher progesterone levels, than controls. These types of analyses could be investigated with the current data set.

A limitation of the findings in this article is that there are several case definitions of ME/CFS as well as long COVID, and this will make it more difficult to study comparable samples across different laboratories. While follow-up data is being collected on patients with ME/CFS, there is also a need to investigate the longer-term consequences of infection with SARS-CoV-2. In addition, our review focused on a prospective study of IM, but there are many other viral infections such as Dengue, West Nile, Chikungunya, etc. The research community would benefit from research on these typeillnesses.

Case definitions are crucial for science, and even more critical for diseases like ME/CFS and long COVID that lack a consistent biomarker. There is a clear benefit for a more uniform case definition because, currently, physicians often make a diagnosis on a case-by-case basis with a mix of definitions and their judgment. However, there are potential negative consequences of overly broad criteria. For example, the National Academies of Sciences, Engineering, and Medicine have recently proposed a new case definition for long COVID [30]. The criteria specify that the condition “is present for at least 3 months as a continuous, relapsing and remitting, or progressive disease state” and the condition can be defined by “single or multiple symptoms” that “can range from mild to severe.” However, a person can meet these proposed long COVID criteria by merely having one symptom that is not a burden to the person or does not have any negative impact on the person’s functioning. If a person has trivial pain in the toe for 3 months following COVID-19 infection, with no negative consequences to the person’s functioning or quality of life, that person would still be eligible for a long COVID diagnosis. The failure to list any thresholds of frequency or severity of symptoms, so that the symptoms are not trivial, has major consequences for an infection that is as widespread as COVID.

The behavioral and pathophysiological underpinnings of both ME/CFS following IM and long COVID are still poorly understood [58]. Prospective longitudinal studies can help in the understanding of post-infection illness following the onset of EBV and SARS-CoV-2. Our ongoing prospective longitudinal study of ME/CFS following IM will hopefully continue to uncover immunologic or metabolomic commonalities and differences between ME/CFS and long COVID.

## Figures and Tables

**Figure 1 microorganisms-13-00702-f001:**
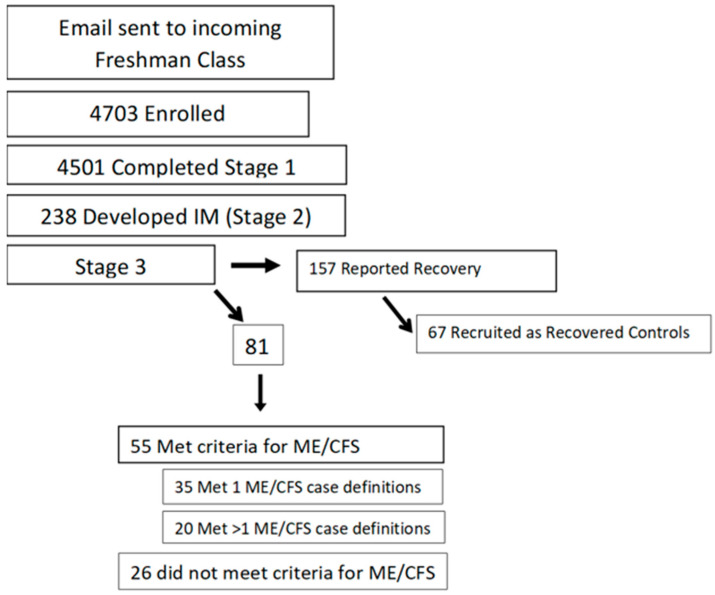
Recruitment of participants for prospective study at Waves 1, 2, and 3.

**Figure 2 microorganisms-13-00702-f002:**
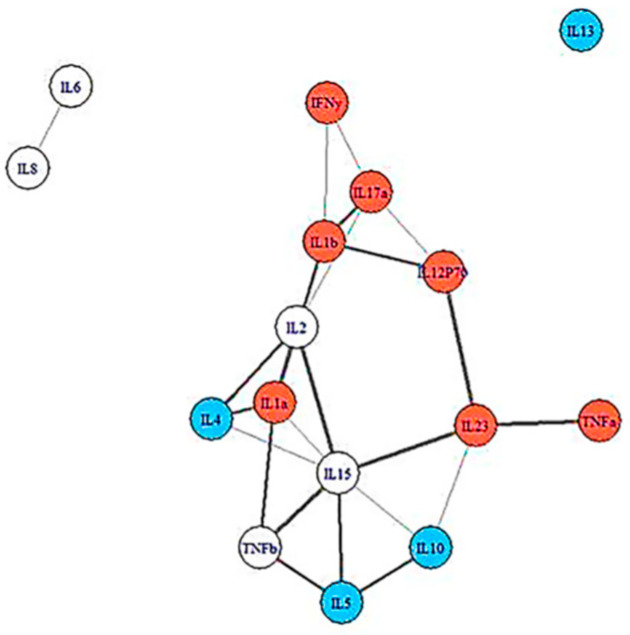
Cytokine data at baseline for participants who eventually had severe ME/CFS. Cytokines colored red are pro-inflammatory, those colored blue are anti-inflammatory. Cytokines not colored have multi-purpose functions.

**Figure 3 microorganisms-13-00702-f003:**
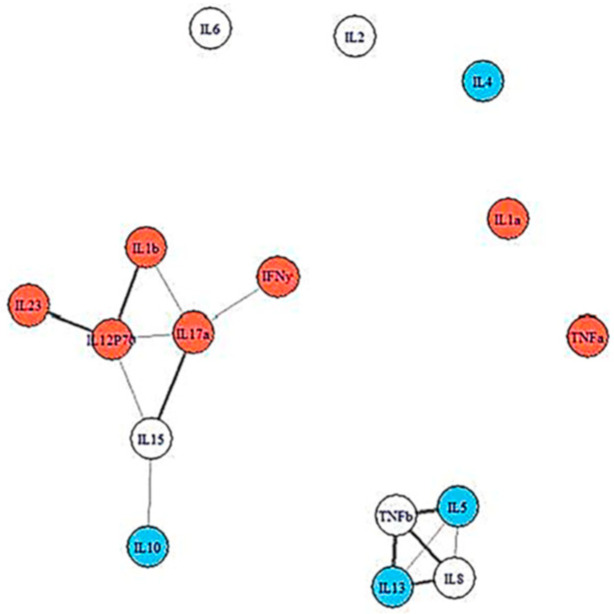
Cytokine data at baseline for Controls. Cytokines colored red are pro-inflammatory, those colored blue are anti-inflammatory. Cytokines not colored have multi-purpose functions.

**Table 1 microorganisms-13-00702-t001:** Significant metabolomic results at baseline (prior to IM) in controls who recovered from IM vs participants who went on to develop severe ME/CFS 6 months following IM.

			S-CFS	Controls		
	KEEG	Metabolite	M(SD)	M(SD)	*U*	*p*
^a^	C00750	spermine	18.79 (0.13)	19.54 (0.16)	0	0.0000000002
^c^	C00354…C00665	F-1,6/2,6-DP	14.94 (0.43)	13.90 (0.24)	321	0.0000000015
^a^	C00315	spermidine	19.18 (0.66)	17.84 (0.33)	311	0.0000000822
^c^	C00002…C00286	ATP/dGTP	14.57 (1.02)	13.16 (0.54)	301	0.0000012586
^b^	C00169	carbamoyl phosphate	15.67 (0.45)	16.65 (0.12)	23	0.0000012586
^a^	C00127	glutathione disulfide	13.63 (1.00)	11.64 (1.11)	300	0.0000015973
^c^	C00158…C00311	citrate/citrate(iso)	22.90 (0.28)	22.40 (0.30)	290	0.0000135233
^a^	C00112	CDP	10.84 (1.29)	8.88 (0.85)	297	0.0000169794

Note: ^a^ = identity confirmed, ^b^ = identity not confirmed, ^c^ = cannot separate. Note: Bonferroni correction (*p* < 0.000038).

## Data Availability

No new data were created or analyzed in this study. Data sharing is not applicable to this article.

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
