# Peer review of "Predisposing and Precipitating Factors in Epstein–Barr Virus-Caused Myalgic Encephalomyelitis/Chronic Fatigue Syndrome"

_microorganisms, 2025, doi:10.3390/microorganisms13040702_

Round 1
Reviewer 1 Report
Comments and Suggestions for Authors
Jason et al. summarized in this manuscript the importance of investigating post-viral syndromes. This topic is relevant to the infectious diseases field. However, the title is a little misleading, as the authors focus only on Myalgic Encephalomyelitis/Chronic Fatigue Syndrome after infectious mononucleosis caused by EBV. As a major comment, this review would benefit from a more extensive revision of the topic including the research done on other post-acute infection or post-viral syndromes and not only on infectious mononucleosis and expanding in more detail the research done in SARS-CoV-2 infection and other viral infections such dengue, west nile, chikungunya, etc. The authors should also evaluate if they would like to include information on other post-acute infection syndromes associated to bacterial or parasitic infections. The research community would benefit from a broader and more inclusive review on this topic of research. As minor comments, there are several typos along the manuscript that should be corrected, the authors should carefully review the text and correct the errors. Furthermore, the figures should be improved and make them more understandable, and the review also would benefit from a graphical summary of the state of the art of post-viral syndromes and/or post-acute infection syndromes. Finally, the authors should use the reference format of the journal and increase the number of references on this topic from other expert groups.
Author Response
Reviewer 1 felt that the title is a little misleading, as we focus only on Myalgic Encephalomyelitis/Chronic Fatigue Syndrome after infectious mononucleosis caused by EBV. We have now changed the title to reflect the focus of our article.
Reviewer 1 also felt that research done on other post-acute infection or post-viral syndromes and not only on infectious mononucleosis and expanding in more detail the research done in SARS-CoV-2 infection and other viral infections such dengue, west nile, chikungunya, etc. We agree that the research community would benefit from a broader and more inclusive review on this topic of research, however, the topic of our article is on a particular research project and not a review of all post-infectious illnesses, and such a review is beyond the scope of our more limited and targeted focus here. The first author has recently published a book that deals with Long COVID and other illnesses, so we are familiar with this literature, but such an article would cover much more territory and as we said, good reviews of this area are elsewhere, and this is mentioned in the text. There are 58 references in this paper, so just trying to focus on our limited objectives is about all we can do; otherwise we would need to write a monograph or a book.
We have now fixed typos in the manuscript.
The figures have now been improved and they are more understandable.
We now used the reference format of the journal and have increased the number of references on this topic from other expert groups.
Reviewer 2 Report
Comments and Suggestions for Authors
Leonard A Jason and Ben Katz's manuscript is entitled "Predisposing and Precipitating Factors in Post-Viral Syndromes". It is an interesting topic, but I have some concerns.
- The title refers to Predisposing and Precipitating Factors in Post-Viral Syndromes, but the content of the manuscript is mostly related to EB virus infections, so it is suggested to change the title.
- The writing style is confusing, as if it were an original work.
- The abstract is not very clear; make sure it is organised. A review abstract is an abbreviated version of the article and should contain all the information necessary for the reader to determine: (1) the problem statement, what the objectives of the study were; (2) how the study was conducted; (3) what the conclusions are and what the significance of the results are.
- Authors should use a single term IM or glandular fever.
- L100.-Authors should summarize Stage 3 criteria here
- Explain how you carried out the network analysis in Figure 2,
- While the review mentions neurological symptoms and some studies on brain function, it could go deeper into the neurological mechanisms underlying post-viral syndromes.
- Although the review mentions a heritable component to ME/CFS, it does not provide a comprehensive overview of the current understanding of genetic risk factors.
- Explain how the authors obtained the results in Table 1.
- Authors should provide current treatment approaches and management strategies for these conditions.
- Authors must indicate the limitations of their work.
Author Response
As with Reviewer 1, Reviewer 2 also felt that the title related to EB virus infections, so we have now changed the title.
In the article, we have now made it clear that this is a review article rather than an original work.
Reviewer 2 felt that the abstract should be organized to include the objectives of the study, how the study was conducted and what the conclusions are and their significance. This would work for a classic study, but our paper is one of a review of a longitudinal series of papers on one data set, and this is now more clearly indicated in the revised abstract.
We now use the term IM rather than glandular fever.
We now provide more information about Stage 3 criteria in the overall study.
We now try to better explain the network analysis in Figure 2, whereas the details of how the analyses were conducted are somewhat complicated and we refer readers to the publication that provides this information.
Reviewer 2 asked us to go deeper into the neurological mechanisms underlying post-viral syndromes, but this is not the focus of our article but we do add some studies on this area for readers with interests in this area. With 58 references now, this paper already covers lots of ground, and we just want to provide some boundaries of what we can review in this paper.
We agree that our article mentions a heritable component to ME/CFS, it does not provide a comprehensive overview of the current understanding of genetic risk factors. But no one paper can cover all areas, as we indicate in the discussion section.
Regarding how we obtained the results in Table 1, there is an article that is referred to that provides descriptions on these biomarkers and other matters, which readers can find, so this info is available elsewhere. Here we just summarize the main findings of that study.
Reviewer 2 suggested that we provide current treatment approaches and management strategies for these conditions. This is a bit beyond the scope of our study, but we now do provide references for this for the readers who might be interested.
We now indicate more on the limitations of our work.
Round 2
Reviewer 1 Report
Comments and Suggestions for Authors
I thank the authors for addressing most of the comments on my previous review. However, I would suggest maintaining the title “Introduction” or “Background” instead of changing it to “Predisposing and Precipitating Factors in Post-Viral Syndromes” which was the title of the review in the previous version. It might be misleading to use that title in any of the sections because the authors focus on Myalgic Encephalomyelitis/Chronic Fatigue Syndrome (ME/CFS) and not post-viral syndromes in general.
Author Response
We agree that the title was a bit too broad so it has now been changed to reflect the focus on ME/CFS.
Reviewer 2 Report
Comments and Suggestions for Authors
This is a review article that presents results without including essential sections like hypotheses, selection criteria, materials, and methods.
Author Response
We agree that this is a review article, but given that there are multiple studies with different methods and materials, it would not be possible to structure the paper like it was one more traditional study. This paper describes multiple and separate studies, and each is summarized in a section, and to put all the methods and stats in sections unto themselves would be very difficult to follow for the readers.